# Diversity of Strategies for Motivation in Learning (DSML)—A New Measure for Measuring Student Academic Motivation

**DOI:** 10.3390/bs13040301

**Published:** 2023-04-01

**Authors:** Caroline Hands, Maria Limniou

**Affiliations:** Department of Psychology, University of Liverpool, Liverpool L69 7ZA, UK; mlimniou@liverpool.ac.uk

**Keywords:** self-report measure, self-regulated learning, self-efficacy, test anxiety, motivation, university students, MSLQ

## Abstract

Although the Motivated Strategies for Learning Questionnaire (MSLQ) has been widely used to measure student motivation, researchers have raised questions regarding its length and several problematic statements. This study introduces a new questionnaire, adapting items from the MSLQ and including three new key themes of course utility, procrastination and use of diverse sources. A total of 1246 students from a university in the northwest of England, studying a range of subjects and from across all grade boundaries, fully completed the questionnaire. Factor analysis suggested a 24-item questionnaire, including 6 factors: test anxiety, self-efficacy, source diversity, study skills, self-regulation and course utility. The measure, Diversity of Strategies for Motivation in Learning (DSML), has good predictive power for students with or without academic successes, and it can be used as a quick and an early alert monitoring tool to measure student motivation and study skills. The DSML has supported various interventions; however, further testing is required in other cultures, languages and educational environments (such as schools and colleges).

## 1. Introduction

In a wide range of academic domains, the success of students depends heavily on their ability to envision, manipulate and navigate complex multidimensional information presented within their studies [1]. When investigating student motivations and behaviors, there is a vast array of methods, measures and interventions from which to choose. Additionally, educators conduct most studies alongside teaching and therefore seek methods that are time-efficient and easy-to-use, -administer and -analyze. Hence, the self-report measure remains one of the most popular choices as this needs minimal input from the researcher to gather data; thus, research should directly focus on improving these measures [2,3].

Despite drawbacks to their use, [4] self-report measures remain the most popular method of data collection and aim to tap into a range of underlying concepts. Such concepts include self-efficacy [5]; learning approach [6]; and self-regulation [7], which have all been shown empirically to have a moderating effect on student outcomes [8,9,10,11]. Common measures used to study this phenomenon include the Revised Two Factor Study Process Questionnaire (RSPQ) [12], the Metacognitive Awareness Index (MAI) [13] and the Motivated Strategies for Learning Questionnaire (MSLQ) [14]. Since its development, the MSLQ has been used across various fields and types of education [15,16,17] and has been cited almost 6700 times, demonstrating its popularity in the field [18]. The measure is comprised of 81 seven-point Likert scale items, measuring student behavior across 15 different subscales. Scores are then summed to produce a single score predicting student study motivations. The MSLQ consists of two primary scales—motivation and learning strategies. The motivation scale is broken down into 6 subscales of 31 items regarding goal beliefs, skills and anxiety related to tests [19]. The learning strategies scale is based on 9 subscales, with 50 items assessing cognitive strategies and resource management skills [20].

Despite its popularity, several issues with the measure have been identified by various scholars. Credé and Phillips’ [21] meta-analysis on the MSLQ reviewed 67 studies covering over 19,000 students and found that the MSLQ offered a large variation in its predictive ability with different subscales ranging from an effect size of 0.4 (effort regulation) to 0.05 (help-seeking). Furthermore, the authors identified items including conditional content statements (e.g., whenever X occurs, I do Y) as being prone to issues regarding their interpretation and clarity [22]. Similarly, ideal point items carry a similar critique in which interpretation and response vary depending on the student’s successes [23]. For example, the item “I ask the instructor to clarify concepts I don’t understand well” is likely to be answered positively by middle-performing students and negatively by both high- and low-performing students, albeit for different reasons. High-performing students would not need clarification while low-performing students would not seek help due to either not realizing they had misunderstood or not bothering to clarify unclear points. 

Another substantial critique of MSLQ is its incorrect assumption that students are heterogeneous across courses and institutions [24]. Students are, in fact, remarkably diverse, displaying variation across their outcome grades, motivation and learning approaches. Through the measure’s transferability (i.e., its use across course types), distinct nuance and course specific factors are not accounted for, which may play a significant part in student motivation and behavior. Additionally, factors, such as one’s psychological state, social networks, support, environment/contextual setting and SES background, will also have a significant relationship (whether that be main or mediating) on student motivation. For example, the disparity between lower and higher SES backgrounds on academic student achievement is well-documented [25], with higher SES students often attaining higher grades through having access to a wider range of higher-quality schools and additional resources and support (e.g., private tutors). As such, Winne and Baker [26] suggested that multilevel cross-validation (a method for assessing the effectiveness of prediction models that involves frequent training and model testing across data subsets) is necessary to combat the above critique before the use of existing self-reporting measures. Issues with the MSLQ’s factor structure have also been noted, leading Dunn et al. [27] to suggest the reconsideration and restructuring of the metacognitive, self-regulation and effort regulation items. Hilpert and colleagues [28] additionally proposed that the extrinsic goal orientation items should be re-examined due to their ambiguity with several other researchers supportive of this point [20,29,30].

It is worth noting that Pintrich’s [31] original MSLQ measure was based on a single sample, and following a confirmatory factor analysis conducted by Muis et al. [32], it has been suggested that the factor structure was not as stable as initially suggested. Upon this finding, it is suggested that the use findings from the MSLQ be interpreted with care if indeed used. This warning point was also echoed by Gable [33] who suggested that diverse samples of students were necessary to establish the validity of the factor structure. 

Additionally, another area of the critique of the MSLQ is related to the current diversity of educational methods and settings (i.e., the utilization of a range of online tools by contemporary students and/or on-campus teaching), while Pintrich’s [31] original student sample was confined to traditional teaching methods and customs used in the 1980s and 90s. Specifically, in the past 30 years, shifts in both teaching practices and the use of technology have vastly changed education [34]. Teaching practices have developed considerably with more use of student-centered teaching practices and an increasing emphasis on the value of formative assessments [27,35]. Similarly, technological modifications have changed how students access and use a range of learning materials, e.g., accessing online journal articles rather than visiting libraries. This multifaceted and ever-increasing role of technology in education makes it inevitable that many of the original MSLQ questions have diminished in their relevancy. Indeed, when Cho and Summers [23] conducted an analysis of the MSLQ in an online learning environment, they found it to be a poor fit, with many of the items no longer appropriate to student learning experiences. The authors suggested that current researchers consider adapting the measure to better fit current students’ situations. These changes within the educational sector make it inevitable that some MSLQ items need adapting to reflect contemporary student learning conditions.

As well as implementing the changes seen in current study methods, the measure’s theoretical basis should also be examined. Many learning inventories are developed top-down using theoretical constructs from cognitive psychology; so, as our knowledge of these constructs develops, so should the methodologies for researching learning approaches [12,36]. For example, deep learning strategies include effort regulation, time management and metacognitive ability, all of which have been shown to strongly correlate with student grade point average (GPA) [37,38,39]. Equally, negative correlations between study habits and grades have been found due to test anxiety [40,41], boredom [42] and procrastination [43,44,45]. While the original MSLQ manual provides correlations with student grades, it unfortunately does not specify which of these were significant.

In addition to these issues in question, it is also important to consider the issue of data collection/the questionnaire structure itself. The full MSLQ contains 15 separate subscales and a total of 81 questions, taking between 20 and 30 min to administer. This can lead to respondents potentially developing survey fatigue [46], responding without reading the question properly or even dropping out of the study, resulting in a reduction in the quality of the data [47]. The lengthier the inventory, both theoretically and practically, the higher the respondent burden. By increasing the time and effort required to complete a measure, response rates and data quality lessen due to students being less likely to provide thoughtful and considered responses. The resultant, and anticipated, poor quality data will likely make staff reluctant to use the MSLQ due to resource and complexity reasons [48]. Conversely, shorter measures use less time and resources and offer increased flexibility for inclusion in larger surveys, or interventions, allowing researchers to adapt their data collection strategy to the needs of their study. While it is possible to use the MSLQ’s subscales individually, confusion as to which subscale is most relevant to the researcher arises, creating a trade-off between resource and psychometric quality [49]. Having said this, scale validity and reliability do not necessarily increase proportionately alongside the increase in items [50]. 

### The Current Study

A pilot study conducted by the current authors running the original MSLQ on 181 University of Liverpool students found that both peer learning and the help-seeking scale had low alpha coefficients and poor internal consistency. These results were found to be consistent with both the measure’s authors and the results from more contemporary scholars [23]. There were also several further issues with the data, such as weak predictive power and extensive evidence of survey fatigue (such as careless and incomplete responses). Therefore, this study aims to utilize previous research to develop a shorter and more focused questionnaire on student motivation. Following the previous studies and literature discussed recommendations, the objective of this study is to create an accurate questionnaire, targeting important but currently missing theoretical, conceptual elements while simultaneously being quicker and easier to administer than the original measure in order to reduce questionnaire fatigue [29]. Another objective of this study is to develop a revised shortened questionnaire suitable for use in modern educational environments that blended learning offers. Finally, this study’s objective is to include items that provide a more holistic view of student learning, such as the surface learning approach [6], self-efficacy [5] and self-regulation [7].

## 2. Methodology

### 2.1. Questionnaire Development

Based on the findings and critiques discussed above, the initial step taken in developing a new measure of student behavior was to examine each of the original MSLQ items to establish which should be retained and which had poor discriminate values [51]. As mentioned above, help-seeking and peer support scales showed a lack of consistency and were removed completely in this initial step; the remaining 74 questions were then individually reviewed. Any question which contained more than one concept was reworded and simplified. For example, question 16: “In a class like this, I prefer course material that arouses my curiosity even if it is difficult to learn” was reworded to “I like material that really challenges me even if it is difficult to learn”. Further questions were also eliminated on an individual basis when it was not possible to re-word, for example, question 31: “Considering the difficulty of this course, the teacher, and my skills, I think I will do well in this class”. This question asks students to weigh out three different things and then decide on a suitable option—something which will inevitably lead to different interpretations between different students; so, it was therefore deleted. Finally, some questions were combined such as question 59: “I memorize keywords to remind me of important concepts in this class” and question 72 “I make lists of important items for this course and memories the list” into “I make lists of important terms or keywords for the course and memorize them”. This process resulted in retaining 46 questions from the original measure, with 7 containing the original wording (as they were shown to have good discriminative validation in the pilot study), 8 with minor word modifications and 31 questions with amended wording. The decision to also include six questions from other measures was informed by previous findings from the study’s author [34] and the highlighted critique aforementioned in the literature review. The six questions were one from Schraw and Denison’s [13] metacognitive awareness index (MAI) and three from Briggs and colleagues’ [12] revised study process questionnaire (R2F-SPQ). As suggested by Credé and Phillips [21], and alongside our own insights, some additional questions were written to address aspects of student motivations and behaviors that were not considered within the original measure. These included questions covering concepts, such as procrastination, locus of control and student use of digital materials. The process of removing and rewording the MSLQ items, along with adding additional questions from other measures, led to the creation of the novel measure—strategies for motivation in learning DSML. 

This initial iteration was then subject to five rounds of revision and review by four educational psychology academics (see acknowledgements). These experts were invited to review the wording of the questions, suggest any potential rewording, identify semantic changes in the question meaning, and point out any errors. This process resulted in some further wording changes and question adjustments to ensure they were sufficiently discriminative to identify differences in student approach. These steps resulted in a final 64-item measure suitable for exploratory factor analysis.

In addition to the 64 questions mentioned above, two free text boxes were provided for student feedback regarding question clarity and any missed areas of assessing their learning behaviors. Students were given the option to either use a dummy or their real ID, if comfortable, to enable the linking of responses to student grades. Finally, the questionnaire asked five demographic questions on one’s predicted grade boundary, year of study, the student’s affiliated department, location of home country and sex. Two further questions (see Appendix A) were included to check the validity of the answers given. In instances where the items referred to paper-based materials, these were reworded to include the appropriate digital aspects. Additionally, some terminology varied across different departments within the University; therefore, minor adjustments were made to the wording to make this clear to the students, for example, including the terms tutor or academic advisor. The finalized questionnaire (see Appendix A) received ethical approval from the University’s ethical board. 

### 2.2. Participants and Procedure

The size of the study’s sample was based on an estimate of around 10–15 students per item, determined by the recommendations of Comery and Lee [52] and Fabrigar et al. [53]. This value was chosen to ensure that both the Exploratory Factor Analysis (EFA) and Confirmatory Factor Analysis (CFA) had sufficient power to assess a robust model of the DSML. The University of Liverpool’s Research Ethics Committee approved this study. Data was collected in April and May of 2018, using a mixture of opportunity and snowball sampling at the University. Recruitment took place in a range of academic and study spaces across the campus (including but not limited to lectures, seminars, the library, etc.). The participants could complete the questionnaire either online or on paper. A participant information sheet was provided to the students followed by a consent form. Only if students agreed to participate in the study was the questionnaire (see Appendix A) then administered (with an average response time of 15 min). Finally, students were debriefed and provided a £1 shopping voucher for their participation

Initially, 1246 responses were collected; 20 responses were incomplete or contained improper responses (e.g., rating all items the same), so they were deleted, leaving 1126 fully completed questionnaires. The data for the initial measure was gathered in several ways. First-year psychology students were invited to complete the questionnaire for course credits (obtaining 264 respondents); second-year students also collected data as part of a course project obtaining 96 responses (these were recruited from second and third-year psychology students); and 77 students from life sciences were recruited following an announcement across two lectures. Finally, the bulk of responses (689) were collected through opportunity sampling across a variety of locations on campus. Faculty breakdown is slightly overrepresented by the School of Psychology and the Faculty of Humanity and Social Sciences, due to data collection locations being based on these corresponding Faculty’s campuses (see Table 1). 

The breakdown of years and grades was representative of the University’s population (as shown in Table 2 and Table 3). Most participants were students from the UK (91.3%), with 4.5% from Europe, 2.2% from Asia and 2% from elsewhere in the world. The sample predominantly identified as female (70.4%), with 24.8% identifying as male, 0.7% identifying as other, 2.8% preferring not to answer and 1.5% providing no response.

From the initial 1126 questionnaires, 124 participants were removed for an assortment of reasons: 8 missed both validation questions; 77 answered incorrectly on question 41 “For this question please select: Not at all true of me”; and 39 gave a rating of three or below for question 66 “My answers are a fair reflection of my true feelings”. In turn, the final number of responses used was 1021.

## 3. Results

The factor analysis approach method has been widely used to evaluate relationships with visible variables or a set of factors by measuring an item or question. Factor analysis involves a series of statistical analyses that employ a similar and functional method instead of a single statistical method [54]. There are two main types of factor analysis: Exploratory Factor Analysis (EFA) and Confirmatory Factor Analysis (CFA). Both aim to create relationships observed in groups composed of a small number of members with only a few hidden variables. However, EFA and CFA often vary in terms of the number and type of instructions and the size of the hidden variables [55]. EFA is frequently utilized in the early phases of scale development and construct validation, while CFA is implemented in later phases when the underlying structure has been established based on empirical and theoretical grounds [55]. As the aim of this study was to develop a new questionnaire for student motivation, EFA was implemented to test the validation of the questionnaire, whilst CFA was employed to establish the theoretical factors. Since structural equation modelling is based on the significance of differences in the covariance matrix, Yeşilyurt [56] suggested that the number of participants should exceed the number needed for each entry in the matrix when such models are built. Participant responses were then randomly allocated into two groups—559 responses were used for the EFA, and 462 responses were used for the CFA. The uneven splitting of the groups was justified due to missing data in the not Missing Completely At Random (MCAR) responses being deemed sufficiently large and representative of the population of interest. The data were analyzed with R 3.5.2 (published by the R project December, 2018) using the Lavaan, Tidyverse and Psych packages. Data were analyzed using an unweighted least squares (ULS) regression. Due to the ordinal structure of the data and our consideration to not make assumptions about the item distributions, this method of analysis was chosen. ULS methodology is best employed when the variances of observed variables are similar [57]. As it was likely that the items in the questionnaire were correlated, an oblimin rotation was used [58].

### 3.1. Exploratory Factor Analysis Results (N = 559)

The primary goal of EFA is to arrive at a more concise and conceptual understanding of one’s set of measured variables [59]. This is determined by the number and nature of common factors required to fit the pattern of correlation among the observed variables [54].

Firstly, the Kaiser-Meyer-Olkin (KMO) measure of sampling adequacy for the initial exploratory factor analysis were run, producing results ranging from 0.81 to 0.88. A value closer to 1 indicates the patterns of correlations are compact, and therefore, factor analysis should yield distinct and reliable factors—producing hopeful findings for our data. Next, we ran Bartlett’s test of Sphericity to check the correlations between the variables. At all steps, the test was significant indicating that the correlation matrix significantly differed from an identity matrix denoting significant correlations between some of the variables within the measure—meeting this factor analysis prerequisite. According to Hays et al., [60] factor loadings that exceed 0.40 are generally considered meaningful. We deleted any items with absolute values greater than 0.35 on more than one factor and any discrepancies between cross-loadings with an absolute value of less than 1.5. Each factor also had to have a minimum of three items [61]. In order to evaluate model-fit we used the criteria recommended by Hu & Bentler [62] suggesting a comparative fit index (CFI) and the Tucker Lewis index (TLI) score of being greater than 0.90 and 0.95, respectively, for good and great fittings models. We also used a root mean square error approximation (RMSA) of less than 0.60 as indicating good model fit. Data was examined through several iterations with redundant items removed at each step (see Table 4).

In total, 40 items were removed from the scale due to either not or ambiguously loading (i.e., not loading strongly) onto any factor. The final model demonstrated a 6-factor good fit solution, with eigenvalues ranging from 2.38 (factor 1) to 1.45 (factor 5). Table 5 outlines the final model’s fit statistics, whilst Table 6 illustrates the item loadings and variance percentages explained by the model.

### 3.2. Confirmatory Factor Analysis Results (n = 461)

While EFA is not based on apriori theory, CFA is, and thus is typically driven by theoretical expectations regarding the structure of the data [63]. The focus of CFA is on how well the measurement model, which operationalizes the theoretical factor structure, fits the empirical data derived from the questionnaire responses. This is frequently assessed using absolute indices, such as the CFI or TLI. 

CFA was run using the Lavaan package in R, with unweighted least squares regression models being the most appropriate model type to use due to the data’s ordinal nature and the apparent clustering seen around point 4 on the 7-point Likert response scales. The CFA model used the factors derived from the EFA, with the model showing a good fit (see Table 7) and explained 47% of the model variance. 

Figure 1 shows the factor loadings for each item and the inter-correlations between the factors; the strongest loadings are apparent for test anxiety, source diversity and course utility.

### 3.3. CFA Results and Grade Boundaries

As well as determining the factor structure, we also explored its effects on student grade boundaries. Note, the data collected on students’ self-reported grade boundaries (measured on a 5-point Likert scale) showed a moderate correlation with the overall grade for the subset of Psychology students, of whom the only overall grades were available for (rs (342) 0.573, *p* = 0.003). This result suggests that, within Psychology, students self-reported grades were a good proxy for student performance. 

As Table 8 shows, the measure’s factors not only correlated strongly to each other but also to student grades. 

## 4. Discussion

The current study aimed to develop a short measure examining student motivations to support blended learning along with statements of higher clarity compared to the MSLQ. As such, the Diversity of Strategies for Motivation in Learning (DSML) questionnaire consists of six factors measuring self-regulation, self-efficacy, source diversity, study strategies, test anxiety and course utility. This structure was affirmed by CFA following the EFA analysis. The final resulting measure contained 3 questions that are unchanged from the original MSLQ; 4 that were subject to minor wording changes; 14 that were based on the original measure but completely reworded; and lastly, 3 newly developed questions (all of which loaded onto the self-regulation factor). 

Both self-regulation and self-efficacy are key performance factors and are linked to successful outcomes [37]. Self-efficacy tends to become less helpful at explaining variations in grades as a course progresses—a finding that is particularly pronounced in lower-performing students [64]. On the other hand, high-achieving students tend to increase their levels of self-efficacy, further improving performance by reinforcing helpful study strategies [65]. It is suggested that the divergence in both self-efficacy and self-regulation could be due to lower-achieving students overestimating their abilities [66]. Furthermore, when combined with a lack of metacognitive abilities, it suggests these students are less likely to learn from previous experiences and to use this to regulate their behavior [67]. Successful self-regulation, self-efficacy and metacognitive ability are typically mediated through students’ learning behaviors [68]. It has been suggested that test anxiety mediates the relationship between self-efficacy/self-regulation and study behaviors [14]. 

Study strategies (i.e., surface learning) and source diversity are interrelated as they reflect how students approach their studies and demonstrate their regulatory and efficacy skills; in addition, source diversity aims to recognize students who learn strategically and possess these skills. Students’ behavioral self-regulation is also worthy of close attention due to the distinctiveness of this learning style [69]. Course utility relates to external motivations (personal, professional and future study) for engaging in studying, offering an explanation for why students engage in particular tasks [19].

Finally, the results of the study indicate that test anxiety is a separate and distinct factor that loads onto its factor, indicating that there is a subgroup of students who experience concerns over exams that are not necessarily related to their study strategies, self-regulation or levels of self-efficacy. This finding highlights the importance of recognizing and addressing test anxiety in students, as it may have a significant impact on their academic performance and overall well-being. By identifying this subgroup of students and providing them with targeted support, educators and mental health professionals can help to alleviate the negative effects of test anxiety and to support these students in achieving academic success. This study aimed to produce a shorter, more focused measure that addressed the key elements of student motivation and performance.

The six key elements of the DSML questionnaire are connected to student engagement and academic performance (Figure 2). Although many researchers have confused motivation with student engagement, student engagement arises from motivation [70,71]. As student engagement is highly related to motivation, the six elements that emerge from this study could be blended with the most widely used student engagement frameworks that support cognitive, affective and behavioral dimensions [72]. The proposed short questionnaire could support studies on student engagement and academic performance, such as a recent study on the COVID-19 pandemic [73].

Taken together, these six factors combine to offer a snapshot of student behaviors and motivations that can be used to assess student behavior demonstrating both one’s “will and skill” [14]. Likely, the cognitive and affective elements (self-efficacy, self-regulation, learning strategies and test anxiety) will be less affected by variations in subject domains [16], while study behaviors (source diversity and study skills) are more likely to be affected by specific situations and contexts [48].

While the proposed DSML questionnaire has addressed some of the issues inherent in the original MSLQ, there are still further areas for improvement. For example, although this measure has been tested in several educational research settings, it has mainly included participants from the same UK University. Outside of the UK, so far the DSML has been only used to support an intervention in the Kingdom of Saudi Arabia [74]; thus, further work on this area is required to explore whether this questionnaire could measure student motivation across both national and international levels. Another limitation is regarding its use in various learning environments. A study conducted during the COVID-19 pandemic has explored the difference in various learning environments by using the DSML questionnaire subscale of study skills to link student motivation and engagement with academic performance [73]. An effective measure will be sensitive to the ability levels of respondents and should reliably measure different populations in a variety of contexts [20]. Thus, future work could further support the validation process of this questionnaire where educational researchers from different countries could test the short DSML measure in various learning environments. As with most self-report measures, a limitation of this measure is that participants need to engage in the processes of question interpretation, relevant event recall and mapping responses onto the scale options [75]. As such, future researchers may also consider using a fully labeled scale (with a description for every point) to reduce the ambiguities in scale interpretation by the participants [76]. In addition, another limitation is related to the level of use that could also influence the validity and reliability of the data collected. For instance, if a participant reports using an approach or strategy infrequently, their responses may not accurately reflect their actual experiences or perceptions. Similarly, if a participant reports using an approach or strategy excessively, their responses may be biased or unreliable. Therefore, it is important for researchers, in the future, to consider the level of the task the measure is addressing and to take account of this when interpreting their findings.

The DSML was developed for use at the course level [68]; however, further testing is needed to see whether this would be suitable for use at the topic or task levels [16,21], which may shed light on some of the score variations across student groups [17,77]. Students have generalized ways of studying that they have reported in line with the current measure; however, as Hardwin et al. [78] point out, learning styles can fluctuate in response to context variations. Attention must also be directed towards further efforts to ensure that future iterations of the measure are culturally suitable for measuring student motivations across a range of students from differing institutions, topics, cultures and languages [20,79,80]. Additionally, it may also be worth investigating the predictive validity of the measure, both in terms of grade prediction and student dropouts. We suggest the measure could be used to identify at-risk students earlier on in their studies [21] by detecting changes in behavior or maladaptive learning strategies [65]. Along with assessing the predictability of the measure at the individual level, we propose that the DSML may be used to test wider disruptions at the societal level, too. For example, testing whether events creating large-scale disruptions to the higher education system, such as the COVID-19 pandemic [34] or national teaching strikes [81], have uniquely driven further changes in student behaviors. The DSML has also been used to test students’ academic performance from three different disciplines when they brought their own devices to a lecture theater [82]. Therefore, the measure should be re-validated to take into account changes in the educational landscape, such as the move to online testing, which has considerably reduced student test anxiety [83,84]. It should also examine whether closely related questions, such as those pertaining to source diversity could be aptly measured by a single item e.g., “I use a variety of sources”, thus further improving the measure’s speed of administration and longitudinal uses to detect changes in students’ motivations. Finally, it is important to note that the currently developed model only explains around one-half of the total variance—a finding likely explained in part by the influence of background factors [2], the measurement of general dispositions rather than actual processes [24,85] and on subjective judgments of one’s own competence [69,86]. In turn, it may be worthwhile for future researchers to triangulate the DSML data with other data sources (both qualitative and quantitative) to establish the stability and fluctuations in behaviors because of such background factors.

The DSML has been developed and designed for university students; however, many of the learning processes it taps into, such as self-regulation and test anxiety, are common to students across education. Therefore, it would be worthwhile testing the DSML in a variety of educational domains such as compulsory schooling, further education and workplace learning. Teachers could use the measure to specifically target interventions for students at risk of disengagement or experiencing test anxiety. Educational researchers could use the measure to assess the effects of structured interventions or unplanned events (such as pandemics and strikes disrupting learning), as well as to measure how these concepts change in students over time.

## 5. Conclusions

In conclusion, the newly proposed DSML measure has been tested for reliability, validity, and uni-dimensionality through both exploratory and confirmatory factor analyses. Research findings confirmed that six factors in three dimensions provided a wide-ranging overview of student thoughts, motivations and behaviors. The 24-item questionnaire provides a valid and reliable measuring scale for universities to utilize to help measure student learning behaviors, predict outcomes and design tailored interventions for low-performing students. In the current environment and landscape of higher education, having indexes measuring and reflecting contemporary student practices is imperative in order to accurately assess modern-day student behaviors and to encourage better overall practices. The development of a valid and reliable measuring scale for student learning behaviors is a crucial step in improving the quality of education and in supporting the success of all students, particularly those who may be struggling. By using this index to identify areas where students may be struggling and providing tailored interventions and support, universities and further education colleges can foster a more supportive and effective learning environment that meets the needs of today’s diverse student population.

## Figures and Tables

**Figure 1 behavsci-13-00301-f001:**
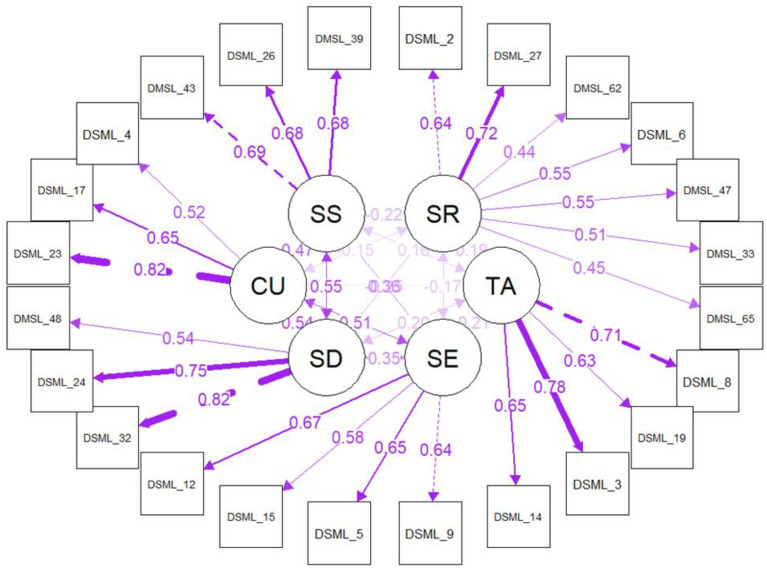
Between and within CFA loadings darker lines represent stronger factor loadings.

**Figure 2 behavsci-13-00301-f002:**
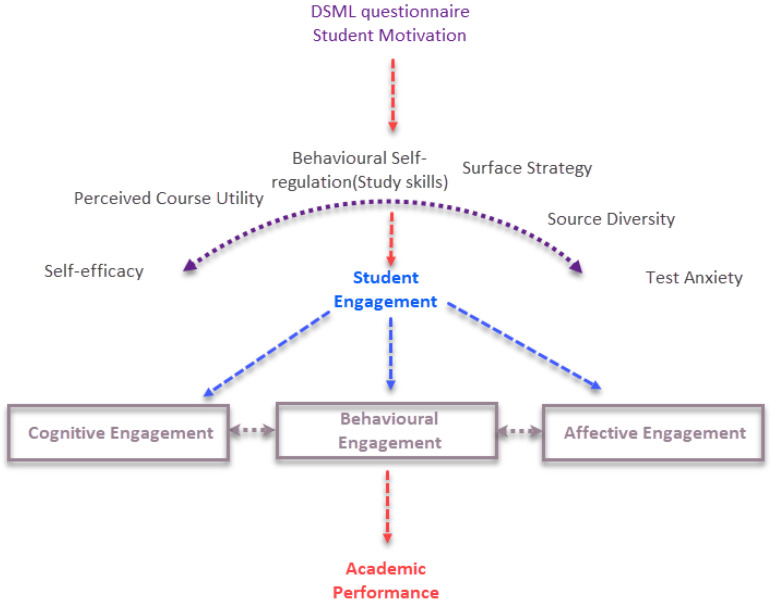
The six component elements of DSML are measuring the three dimensions of the student engagement framework that theoretically predict student performance.

**Table 1 behavsci-13-00301-t001:** Breakdown of responses based on discipline (*N* = 1087).

Topic/Faculty	*n*	Percentage (%)
School of Psychology *	382	33.9
Faculty of Health and Life Sciences	189	16.8
Faculty of Science and Engineering	140	12.6
Faculty of Humanities and Social Sciences	312	27.7
School of Medicine	52	4.6
Other	12	1.1

* Psychology falls under Health and Life sciences but is provided separately as this received the most responses.

**Table 2 behavsci-13-00301-t002:** Breakdown of students’ self-reported predicted grades (*N* = 1111).

Grade	*n*	Percentage (%)
First class (71–100)	149	13.4
2:1 class (60–69)	628	56.5
2:2 class (50–59)	228	20.5
Third class (40–49)	28	2.5
Failing grade (below 40)	3	0.3
Unable to estimate	75	6.8

**Table 3 behavsci-13-00301-t003:** Breakdown of the year of studies (*N* = 1115).

Year of Study	*n*	Percentage (%)
First-year undergraduate	553	49.1
Second-year undergraduate	307	27.3
Third-year and above undergraduate	164	14.7
Post-taught students (Master’s)	73	6.5
Postgraduate research students (PhD)	17	1.5
Postdoc staff	1	0.1

**Table 4 behavsci-13-00301-t004:** EFA steps.

EFA Step	Initial Suggested Factors	KMO	Items Removed
1	11	0.88	28
2	8	0.83	6
3	8	0.83	1
4	8	0.83	1
5	7	0.81	1
6	7	0.82	2
7	6	0.82	1

Note: Bartlett’s test of Sphericity was significant at all steps (*p* > 0.001).

**Table 5 behavsci-13-00301-t005:** Model fit statistics for EFA.

RMSEA	90% CI Lower	90% CI Upper	TLI	BIC	χ^2^	df	*p*
0.045	0.038	0.005	0.916	620.34	4384.66	147	<0.001

**Table 6 behavsci-13-00301-t006:** Final model item loadings and variance scores.

Factor	Contained Items	Model Variance (%)
1. Self-Regulation	7	22
2. Test Anxiety	4	18
3. Self-Efficacy	4	16
4. Source Diversity	3	16
5. Course Utility	3	15
6. Study Strategies	4	13
Total (final model)	25	45

**Table 7 behavsci-13-00301-t007:** Model fit statistics for CFA using standard and robust modelling.

	RMSEA	90% CI Lower	90% CI Upper	TLI	CFI	χ^2^	df	*p*
Standard	0.129	0.124	0.135	0.905	0.918	1808.11	237	<0.001
Robust	0.040	0.033	0.047	0.883	0.900	390.230	237	<0.001

**Table 8 behavsci-13-00301-t008:** Spearman’s correlation matrix shows relationships between factors and self-reported grade boundaries.

Factor (*n*)	M (±SD)	1. Grade	2. Self-Efficacy	3. Self-Regulation	4. Study Skills	5. Test Anxiety	6. Source Diversity	7. Course Utility
1. Grade (945)	3.87 (±0.70)	-	0.445 ** (*n* = 931)	0.199 ** (*n* = 917)	0.730 * (*n* = 992)	−0.105 * (*n* = 888)	0.196 ** (*n* = 939)	0.140 ** (*n* = 934)
2. Self-efficacy (1006)	4.95 (±0.93)		-	0.147 ** (*n* = 977)	0.208 ** (*n* = 992)	−0.158 ** (*n* = 948)	0.273 ** (*n* = 998)	0.367 ** (*n* = 995)
3. Self-regulation (992)	4.47 (±1.09)			-	0.158 ** (*n* = 980)	−0.143 ** (*n* = 937)	0.196 ** (*n* = 985)	0.132 ** (*n* = 979)
4. Study skills (1007)	5.31 (±1.09)				-	0.174 ** (*n* = 950)	0.428 ** (*n* = 1000)	0.339 ** (*n* = 994)
5. Test anxiety (962)	5.09 (±1.20)					-	0.122 * (*n* = 954)	0.069 (*n* = 950)
6. Source diversity (1023)	5.24 (±1.13)						-	0.422 ** (*n* = 1000)
7. Course utility (1008)	5.60 (±1.01)							-

* *p* > 0.005, ** *p* > 0.001.

## Data Availability

All data can be found here: Caroline Hands, & Maria Limniou. (2023). Data set to support the DSML measure [Data set]. Zenodo. https://doi.org/10.5281/zenodo.7789885 (accessed on 27 February 2023).

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
