# Peer review of "Diversity of Strategies for Motivation in Learning (DSML)—A New Measure for Measuring Student Academic Motivation"

_behavsci, 2023, doi:10.3390/bs13040301_

Round 1

Reviewer 1 Report

Dear Editors

Dear authors,

Thank you for the opportunity to read and review this paper. Interesting study about development a new measure for measuring student academic motivation.

Before considering publishing this paper, this paper needs several improvements.

Abstract section:

(Cho and summers 2013, Duncan and McKeachie 2005)… please do not added any citations on abstract section. Just following the structure to writing article abstract, problem background, purposes of study, methodology, findings, implementations, contribution and recommendations.

Initially, 1246 responses were collected, … when? Where? How? The author collect the data questionnaire? How about consent form? Institutional review board? Not clear.

How authors analyze the data? what software? how to interpretate? What is the steps? not clear.

Author need to separate the discussion section and conclusion section

Please add recommendations for further study, recommendation for school, teachers and other researchers.

Please add implication of this study.

Do not forget to follow the MDPI template article.

Author Response

Reviewer 1

The authors would like to thank reviewer 1 for the constructive comments. After careful consideration of these comments, the authors have made the appropriate amendments. The comments are much appreciated, and we have addressed all the points raised.  

Reviewer’s comment:  Abstract section: (Cho and summers 2013, Duncan and McKeachie 2005)… please do not added any citations on abstract section. Just following the structure to writing article abstract, problem background, purposes of study, methodology, findings, implementations, contribution and recommendations.

Authors’ reply: The authors followed the reviewer suggestions and changed the abstract providing a clearer version. However, the abstract word limit cannot allow the authors to provide more detailed points.

Reviewers’ comment:  Initially, 1246 responses were collected, … when? Where? How? The author collect the data questionnaire? How about consent form? Institutional review board? Not clear.  

Authors’ reply: The participants and procedure section has been updated including more details regarding the recruitment process, ethics and consent information. Details of the pilot study have been expanded and moved to the end of the introduction. Additionally all documentation, including Consent and PIS can  be uploaded alongside the raw data for interested readers to access.

Reviewer’s comment:  How authors analyze the data? what software?

 Authors’ reply: The software information was provided in the first paragraph of the results section and it has been highlighted for your reference.  

Reviewer’s comments:   how to interpretate? What is the steps? not clear.

Authors’ reply:  Assuming the reviewer referred to the steps of the exploratory and confirmatory factor analysis, the authors have added an overall explanation at the beginning of the results section and of course they would like to mention that these have been also covered in some depth in the results section. These have been reviewed and a few minor changes made for clarity.

The measure was developed using a factor analysis process. The data was not analysed to a great extent in terms of differences between groups, as the purpose was to develop the new questionnaire for student motivation rather than identify differences between student groups in terms of analysis. Factor analysis (EFA and CFA) has been used to evaluate relationships with visible variables or a set of factors by measuring an item or question. The authors have provided a short discussion at the beginning of the results section about the factor analysis and the difference between EFA and CFA.

The demographic statistics provided are descriptive statistics to help the reader understand the sample. In addition, the correlation matrix  shows the relationship between student grade and the factors of the developed measure; this is not intended to show differences between groups.

Reviewer’s comments: Author need to separate the discussion section and conclusion section

Authors’ reply:  A separate conclusions section has been added and this expands on the previous conclusion.

Reviewer’s comments: Please add recommendations for further study, recommendation for school, teachers and other researchers.

Authors’ reply:  A new paragraph has been added to the end of the discussion, addressing further recommendations for school settings and suggestions for its use by teachers and other researchers.

Reviewer’s comments: Please add implication of this study.

Authors’ reply: Several further implications of the study have been added to the discussion section. Part of the implications is the extent that this questionnaire could support various interventions for student engagement, digital capabilities and academic performance. The additional explanation in figure 2 illustrates how DSML, student engagement and academic performance are linked all together. 

Reviewer’s comments: Do not forget to follow the MDPI template article.

Authors’ reply:   This has now been amended to align with the journal’s template format.

Reviewer 2 Report

This study studied a very interesting topic. The research needs, results, and discussions are very well documented. The research needs, methods, results, and discussions are very well documented.

 Did you develop the questionnaire in research methods? It is judged that the description of the questionnaire development is insufficient. Due to the lack of research methods for questionnaire development, it is difficult to trust the research results.

Prior research and references are different from the journal format.

However, the need for research and discussion is judged to be very good. Due to COVID-19, new research has been attempted in the field of education. 

The conclusion part is relatively lacking. It is necessary to present a summary of this study and its academic implications. Also, please describe your suggestions for follow-up research.

Author Response

Reviewer 2

The authors would like to thank reviewer 2 for the suggestions and for the constructive feedback that they received. After careful consideration of these suggestions, the authors have made the appropriate amendments.

Reviewer’s comments: Did you develop the questionnaire in research methods? It is judged that the description of the questionnaire development is insufficient. Due to the lack of research methods for questionnaire development, it is difficult to trust the research results.

Authors’ reply: Further information relating to the pilot study under the current study and methodology sections. Further detail has been added to the description of the measure development and recruitment processes. A more detailed explanation of the factor analysis process has been included to increase clarity of this methodology.

Reviewer’s comments: Prior research and references are different from the journal format.

Authors’ reply: This has now been amended to align with the journal’s referencing format.

Reviewer’s comments: However, the need for research and discussion is judged to be very good. Due to COVID-19, new research has been attempted in the field of education. 

Authors’ reply: Thank you for this comment. The authors agree new research is necessary as a result of the COVID-19 pandemic. This questionnaire has been included in a study regarding the digital skills which was conducted during the pandemic and the results were consistent with the previous studies. The pandemic has also now been mentioned more specifically in terms of the implications of the study. Additionally the authors are currently planning a follow-up study assessing these effects and re-validating the study in terms of student changes, for example it is necessary to assess changes to levels of test anxiety in students taking online exams. This point has been mentioned in the discussion part as a future work.

Reviewer’s comments: The conclusion part is relatively lacking. It is necessary to present a summary of this study and its academic implications. Also, please describe your suggestions for follow-up  research.

Authors’ reply: The conclusions have been separated and now include a summary of the findings. further suggestions for follow-up research and academic implications have been added to the discussion section, in particular the final paragraph.

Reviewer 3 Report

Paper deals with important topic of student’s motivation in learning and addresses a new approach and instrument in measuring and quantifying motivation in students. Work Is based mostly on evaluation of Motivated Strategies for Learning Questionnaire (MSLQ) as valid and reliable instrument for measuring student-s motivation. However, actual causes and mechanisms of student’s motivation, based on research data, are not explained in sufficient detail.

In order for text to be more approachable and data to be clearer some aspects of the paper may be improved:

-          Short description of the purpose and aim of the research should be mentioned in the abstract.

-          Limitations of the research should be also mentioned in the abstract.

-          Aim, goals and objectives of the research should be explicitly formulated in methodology section. 

-          Hypothesis or research question should be formulated in methodology section or at least some explanation of expected outcomes.

-          Figure 2 should be explained in more detail and in clearer and more comprehensive manner.

-          Separate conclusion section with short recapitulation of key findings would help transparency and clarity of the text.

-          Limitations of the research should be described in methodology section and briefly addressed in conclusion.

-          Future possibilities of measuring student’s motivation in learning with MSLQ should be described and addressed in conclusion.

Author Response

Reviewer 3

The authors would like to thank reviewer 3 for the suggestions and for the constructive feedback that they received. After careful consideration of these suggestions, the authors have made the appropriate amendments.

Reviewer’s comments: Paper deals with important topic of student’s motivation in learning and addresses a new approach and instrument in measuring and quantifying motivation in students. Work Is based mostly on evaluation of Motivated Strategies for Learning Questionnaire (MSLQ) as valid and reliable instrument for measuring student-s motivation. However, actual causes and mechanisms of student’s motivation, based on research data, are not explained in sufficient detail.

Authors’ reply: Figure two has been updated to make these links clearer and motivation is discussed more explicitly in the discussion section. 

Reviewer’s comments: In order for text to be more approachable and data to be clearer some aspects of the paper may be improved:  Short description of the purpose and aim of the research should be mentioned in the abstract. Limitations of the research should be also mentioned in the abstract.

Authors’ reply:  The authors have changed the abstract including making the aim of the research more explicit (the introduction of the new measure with details of its development) and included a limitation of this study, however they cannot further expand the limitations due to the abstract word limit.

Reviewer’s comments: Aim, goals and objectives of the research should be explicitly formulated in methodology section. 

Authors’ reply: The overall aim and the objectives of this study has been clearly provided in the “Current Study” section. 

Reviewer’s comments: Hypothesis or research question should be formulated in methodology section or at least some explanation of expected outcomes.

Authors’ reply: The authors feel that this comment somehow overlaps with the previous one about the objectives and it would be a repetitive process to formulate a research question/hypothesis as the objectives actually represent the research questions.

Reviewer’s comments: Figure 2 should be explained in more detail and in clearer and more comprehensive manner.

Authors’ reply: A more detailed explanation has been provided for the updated figure 2 version, that aims to explain the DSML measure in relation to student engagement and performance.

Reviewer’s comments: Separate conclusion section with short recapitulation of key findings would help transparency and clarity of the text.

Authors’ reply:  The conclusions have been separated and now include a summary of the findings. further suggestions for follow-up research and academic implications have been added to the discussion section, in particular the final paragraph.

Reviewer’s comments: Limitations of the research should be described in methodology section and briefly addressed in conclusion.

Authors’ reply: In the methodology section, the authors have been focused on writing the rationale for the questionnaire development process and presenting the factor analysis. The limitations of the study have been discussed with future directions in the discussion section. .  

Reviewer’s comments: Future possibilities of measuring student’s motivation in learning with MSLQ should be described and addressed in conclusion.

Author’s reply: One of the future directions mentioned in the discussion relates to revalidation of this measure to take account of changes in education as a result of the recent pandemic, for example a move to online exams. We also hope to collect further detail with samples outside of the UK to assess the transferability of the measure.

Reviewer 4 Report

This is a strong and coherent paper that attempts to strengthen the MSLQ instrumentation. Critical aspects are presented to justify the revision of the measures. However, there are some areas for minor improvement that include:

1. While one can speculate, it is important to spell out the research questions or objectives for the paper.

2. More justification on the need for a shorter measure. In doing so, the author can consider elaborating on whether the call for a shortened instrument is within the region where it was developed or outside of this. It is necessary to contextualize and justify this move.

3. Additionally, it may be helpful if there is more information on which aspects of the measure were retained and why? What were the findings of previous studies using this measure? Did they provide any credibility or has this varied within and across contexts? 

4. Method of data collection-We know as readers that the study was based on a particular university. Yet, the limitations of this approach to advance a refined instrument is not presented. More is needed on the limitations of this and on the interpretations of the findings that emerge.  

4. Given the hint that the theoretical foundation of the measure is in need of further examination, then it is important to make related recommendations for advancing this work in the discussion.

Author Response

Reviewer 4  The authors would like to thank reviewer 4 for the suggestions and for the constructive feedback that they received. After careful consideration of these suggestions, the authors have made the appropriate amendments.

Reviewer’s comments: While one can speculate, it is important to spell out the research questions or objectives for the paper.

Authors’ reply: The aim and the objectives of this study have been expanded and clarified and are provided in the “Current Study” section.

Reviewer’s comments: More justification on the need for a shorter measure. In doing so, the author can consider elaborating on whether the call for a shortened instrument is within the region where it was developed or outside of this. It is necessary to contextualize and justify this move.

Authors’ reply: The final paragraph before the current study is introduced has been expanded to provide an increased justification for the use of a shorter measure including student factors (boredom poor response and dropout) and staff factors (increased cost and resources as well as being harder to use as part of a wider measure)

Reviewer’s comments:. It may be helpful if there is more information on which aspects of the measure were retained and why? What were the findings of previous studies using this measure? Did they provide any credibility or has this varied within and across contexts? 

Authors’ reply: The introduction has been updated to discuss the initial pilot study in more detail and further details on the questions retained are provided in the methodology. It’s also possible to see these in the appendices, which lists all questions, plus the level of amendment to them.

Reviewer’s comments:Method of data collection-We know as readers that the study was based on a particular university. Yet, the limitations of this approach to advance a refined instrument is not presented. More is needed on the limitations of this and on the interpretations of the findings that emerge. 

Authors’ reply: The paper has been updated to clarify data collection methods, limitations have been expanded on and discussed in more detail in the discussion section including details of a study the new measure with school children in Saudi Arabia. The authors acknowledge the limitations of collecting data at a single institution, work is currently underway to test this with students being recruited worldwide, as part of a re-validation of the measure following the dramatic changes to the educational landscape as a result of the pandemic.

Reviewer’s comments: Given the hint that the theoretical foundation of the measure is in need of further examination, then it is important to make related recommendations for advancing this work in the discussion.

Authors’ reply: The authors have amended figure 2 to place more emphasis on the theoretical foundations and included several further suggestions of future research direction in the discussion section of this paper.

Round 2

Reviewer 1 Report

Can be accepted in current form

Reviewer 2 Report

This article faithfully followed the revision instructions.

It is necessary to revise the whole by referring to the academic society form.

It is judged to be an articile that can be published by this society.